# Does CytoSorb Interfere with Immunosuppression? A Pharmacokinetic and Functional Evaluation

**DOI:** 10.3390/pharmaceutics17111468

**Published:** 2025-11-13

**Authors:** Stephan Harm, Claudia Schildböck, Denisa Cont, Viktoria Weber, Jens Hartmann

**Affiliations:** Department for Biomedical Research, University for Continuing Education Krems, 3500 Krems, Austria; claudia.schildboeck@donau-uni.ac.at (C.S.); denisa.cont@donau-uni.ac.at (D.C.); viktoria.weber@donau-uni.ac.at (V.W.); jens.hartmann@donau-uni.ac.at (J.H.)

**Keywords:** cytokines, CytoSorb, transplantation, immunosuppressive, hemoadsorption, hemodialysis, protein binding

## Abstract

**Background/Objectives**: Cytokine release during organ transplantation contributes to primary graft dysfunction and requires careful immunomodulation. CytoSorb, a hemoadsorption device developed to reduce circulating cytokine levels, is increasingly used in critically ill patients. However, its impact on concurrent immunosuppressive therapy remains unclear. **Methods**: In this ex vivo study, we investigated the adsorption of five immunosuppressants—cyclosporine A, tacrolimus, methylprednisolone, mycophenolic acid, and 6-mercaptopurine—using a scaled-down CytoSorb hemoadsorption circuit and compared results to chronic and acute dialysis. Additionally, a whole blood model was used to assess the functional impact of CytoSorb treatment on leukocyte activation, using LPS and anti-CD3 stimulation and subsequent cytokine measurement (TNF-α, IL-1β, IL-6, IL-8). **Results**: CytoSorb significantly reduced serum levels of methylprednisolone (92 ± 3%), mycophenolate (80 ± 2%), 6-mercaptopurine (65 ± 32%), and cyclosporine A (61 ± 16%), but had no significant effect on tacrolimus. Dialysis effectively removed methylprednisolone and 6-mercaptopurine, while strongly protein-bound drugs such as cyclosporine A and tacrolimus remained largely unaffected. In the whole blood model, CytoSorb treatment did not significantly alter cytokine release after immunostimulation, suggesting preserved immunosuppressive efficacy. **Conclusions**: CytoSorb treatment reduces the plasma concentration of selected immunosuppressants. However, short-term treatment appears to have minimal impact on immunosuppressive function. These findings support the cautious use of CytoSorb in transplant settings but highlight the need for in vivo confirmation.

## 1. Introduction

Organ transplantation is frequently accompanied by a pronounced inflammatory response resulting from ischemia–reperfusion injury and the release of cytokines, which may trigger a systemic inflammatory response [1]. This hyperinflammatory state contributes to primary graft dysfunction and early allograft injury, thereby jeopardizing transplant outcomes [2]. Conventional immunosuppressive regimens mainly target the adaptive immune system to prevent rejection but do not effectively counteract the initial surge of pro-inflammatory cytokines and damage-associated molecular patterns that occur perioperatively [3].

To address this gap, extracorporeal cytokine adsorption has emerged as a promising adjunctive strategy. The CytoSorb device (CytoSorbents Corporation, Princeton, NJ, USA) is a hemoadsorption cartridge that removes a broad spectrum of middle molecular weight substances, including cytokines, from circulating blood [4,5]. CytoSorb is CE-marked for the removal of middle molecular weight substances—including cytokines—from blood in hyperinflammatory conditions such as septic shock and ARDS [6,7]. However, the translation of cytokine reduction into improved clinical outcomes remains a matter of ongoing debate. Several randomized controlled trials on extracorporeal blood purification approaches, including endotoxin removal using polymyxin-B hemoperfusion (EUPHRATES [8] and ABDOMIX [9]) and cytokine adsorption using CytoSorb (CYCOV [10]), have failed to demonstrate a survival benefit. A recent expert commentary even cautioned against the routine clinical use of hemoadsorption due to the lack of robust patient-centered outcomes [11]. Furthermore, the most recent S3 guideline for sepsis (2025) explicitly recommends that non-selective adsorption devices should not be used routinely in patients with sepsis or septic shock [12]. Thus, while CytoSorb and similar devices have been applied in Europe in clinical practice and in investigational settings, their routine use is not supported by guideline consensus.

These characteristics have prompted its further evaluation in solid organ transplantation. Preclinical studies in kidney, liver, heart, and lung models have shown that CytoSorb can attenuate inflammatory signaling, improve hemodynamics, and reduce tissue injury during organ preservation and reperfusion [13,14,15,16]. Clinically, perioperative use of CytoSorb in high-risk heart and liver transplant recipients has been associated with reduced vasopressor requirements and shorter ICU stays, without compromising graft outcomes [13,14].

A key consideration for the use of CytoSorb in transplant patients is its potential interaction with immunosuppressive drugs. Pharmacokinetic studies in large animal models as well as in vitro systems suggest that CytoSorb has limited impact on blood levels of calcineurin inhibitors such as tacrolimus and cyclosporine, while more hydrophilic agents such as mycophenolate and corticosteroids may be adsorbed to a greater extent [2,6]. However, clinical relevance appears limited during short-term treatment.

The aim of this study was to investigate the effect of CytoSorb treatment and hemodialysis on the blood levels of selected immunosuppressive agents—cyclosporine, tacrolimus, methylprednisolone, mycophenolic acid, and 6-mercaptopurine—using an in vitro whole blood model.

## 2. Materials and Methods

### 2.1. Materials

Acute dialysis was performed using a Fresenius 4008H system with an FX60 High Flux dialyzer (1.4 m^2^ surface area), and chronic dialysis with a multiFiltrate device equipped with an AV600 Ultraflux filter (1.4 m^2^). Both systems were provided by Fresenius Medical Care (Bad Homburg, Germany). CytoSorb cartridges were obtained from CytoSorbents Europe GmbH (Berlin, Germany). Plasma was sourced from the Red Cross Blood Center Linz (Linz, Austria). The immunosuppressants—cyclosporine A (CsA), tacrolimus (TAC), methylprednisolone (MPR), mycophenolic acid (MPA), and 6-mercaptopurine (6-MP)—as well as urea were purchased from Sigma-Aldrich (St. Louis, MO, USA). Immunosuppressants were grouped to mimic clinical polypharmacy: group 1 included CsA and 6-MP while group 2 included MPR, MPA, and TAC.

### 2.2. Human Whole Blood and Plasma

Plasma was obtained from the Red Cross Blood Center Linz (Linz, Austria). Whole blood was collected from healthy donors with informed consent, under the approval of the Ethics Committee of the University for Continuing Education Krems (EK GZ 13/2015-2018). All experiments complied with the Declaration of Helsinki.

### 2.3. Determination of Protein Binding

To assess protein binding of the tested immunosuppressants, human plasma was spiked with the respective drugs and urea as a freely diffusible control. Samples were then subjected to ultrafiltration (30 kDa cutoff, 15,000× *g*, 20 min). The *protein-bound* fraction was calculated using the following formula:(1)$$protein\;binding\;\left[\%\right]=\;1-\frac{C_{filtrate}}{C_{plasma}}$$
where *c_plasma_* is the concentration of each compound in plasma and *c_filtrate_* is its concentration in the ultrafiltrate.

### 2.4. Static CytoSorb Adsorption Setup

Plasma was obtained by centrifugation (3500× *g*, 10 min) from citrate-anticoagulated human whole blood and spiked with immunosuppressants to defined target concentrations (Table 1). To evaluate adsorption, plasma was incubated with the CytoSorb resin at 1:10 (*v*/*v*) for 60 min at 37 °C on a roller mixer. In parallel, a separate aliquot of spiked plasma was incubated with autologous blood cells (1:1, *v*/*v*) to assess cellular uptake of immunosuppressants. Untreated spiked plasma served as control (Figure 1a). Samples were collected before and after incubation for subsequent analysis.

### 2.5. Dynamic CytoSorb Adsorption Setup

To quantify the depletion of selected immunosuppressants during CytoSorb treatment under dynamic conditions, a scaled-down extracorporeal circulation system was established, consisting of a 30 mL CytoSorb adsorption cartridge and a pediatric tubing set filled with 400 mL of freshly drawn human whole blood (Figure 1b). To distinguish between drug removal by adsorption versus uptake by blood components, a control circuit with an identical setup but an empty cartridge was run in parallel for each experiment. Whole blood from two healthy donors of the same blood group was pooled, anticoagulated with citrate (final concentration: 7 mM), spiked with the immunosuppressants, and divided into two 400 mL blood bags. Reflecting typical clinical practice, the immunosuppressants were tested in two representative groups: group 1: cyclosporine A (1200 ng/mL) and 6-mercaptopurine (1000 ng/mL), group 2: methylprednisolone (200 ng/mL), mycophenolate (4000 ng/mL), and tacrolimus (20 ng/mL).

Prior to circulation, the entire system was flushed with isotonic saline. Then, one of the prepared blood bags (Group 1 or Group 2) was connected to the circuit and circulated for 30 min at 37 °C without the adsorber cartridge to allow for equilibration. Following this equilibration, a baseline sample (0 min) was collected, and the adsorption column was introduced into the circuit. In the adsorption circuit, samples were collected at 15, 30, 60, 120, 180 and 240 min. In the control circuit, samples were taken at baseline and after 240 min. All blood samples were centrifuged (3500× *g*, 10 min) to separate plasma, which was stored at −80 °C until further analysis.

### 2.6. Removal of Immunosuppressants by Dialysis

To evaluate the effect of dialysis under clinically relevant conditions, both chronic and acute dialysis settings were simulated using standard Fresenius systems. In all experiments, 1.5 L of citrated human plasma was dialyzed against calcium-free dialysate at 37 °C. Flow parameters were adjusted according to typical clinical use:Chronic dialysis: Fresenius 4008H with FX60 high-flux dialyzer, plasma flow = 140 mL/min (≈200 mL/min blood flow), dialysate flow = 500 mL/min (Figure 1c).Acute dialysis (CVVHD): multiFiltrate system with AV600 ultra-flux dialyzer, plasma flow = 140 mL/min, dialysate flow = 50 mL/min (Figure 1d).

Prior to circulation, the systems were flushed with 2 L of isotonic saline. Plasma samples were collected at defined time points (15–480 min) and analyzed for immunosuppressants, urea, and albumin. Chronic dialysis represents intermittent high-flow conditions, whereas acute dialysis mimics continuous low-flow therapy used in critically ill patients.

### 2.7. Clearance Calculation

The clearance (*C*) of each compound during both the dialysis and the dynamic CytoSorb experiments was calculated using the first-order exponential decay formula:(2)C = Vt  ×lnCoCt
where *V* = plasma volume, *t* = treatment time [min], *C*_0_ = concentration before starting treatment, *C_t_* = concentration at time *t*.

### 2.8. Whole Blood Cell Model

To investigate whether the in vitro depletion of immunosuppressants by CytoSorb affects immunosuppression, a whole blood cell model was established. The experiment was conducted over two consecutive days. A graphical overview of the setup is provided in Appendix A. On day one, freshly drawn human whole blood was incubated overnight at 37 °C with various immunosuppressants, either individually or in combination, to allow for sufficient drug exposure prior to CytoSorb treatment. Since immunosuppressants are typically administered in combination in clinical settings, the substances were grouped based on commonly used therapeutic regimens and distinct mechanisms of action: group 1 included CsA and 6-MP, representing a combination of a calcineurin inhibitor with an antiproliferative agent while group 2 consisted of TAC, MPR, and MPA, combining another calcineurin inhibitor with a corticosteroid and an antimetabolite.

Subsequently, the blood was split into two experimental groups: one was treated with CytoSorb in a batch setup, and the other served as an untreated control, incubated under identical conditions at 37 °C on a roller mixer. Following treatment, all samples were allowed to rest for 4 h. After the resting period, blood samples were stimulated with either lipopolysaccharide (LPS) *E. coli* or with anti-CD3 antibodies. LPS primarily activates monocytes and neutrophils via the TLR4/CD14/MD2 complex, while anti-CD3 specifically stimulates T cells. The stimulation was carried out for 4 h at 37 °C on a roller mixer. Thereafter, the blood samples were centrifuged, and the plasma was collected and stored at −80 °C. Leukocyte activation was assessed by quantification of IL-1β, IL-6, IL-8, and TNF-α.

### 2.9. Quantification of Immunosuppressants and Cytokines

CsA, MPR, and TAC were quantified by ELISA (FK-506, Abnova, Taipei City, China; BlueGene Biotech, Shanghei, China; MaxSignal, Bio Scientific, Austin, TX, USA, respectively). MPA was determined using a validated test for the Cobas c311 autoanalyzer from Roche (Basel, Switzerland). 6-MP was quantified by reversed-phase HPLC [17,18]. After precipitation of the plasma sample with ice cold methanol in ratio 1:5, 30 µL of the supernatant was injected into a Nukleosil 100-5C18, 4.6 × 150 mm C18 column (Macherey-Nagel, Düren, Germany), combined with an Onyx Monolithic C18, 4.6 × 5 mm guard column (Phenomenex, Torrance, CA, USA). The mobile phase consisted of methanol and deionised water (50:50) which was continuously pumped with a flow rate of 0.8 mL/min for five minutes. 6-MP was quantified with a retention time of 2.3 min using an UV detector (325 nm). The quantification of cytokines was determined by Magnetic Luminex Assay with a human premixed Multi Analyte Kit (R&D Systems, Minneapolis, MN, USA). The urea, albumin and total protein levels were measured with the cobas c311 automated analyzer with corresponding test kits from Roche (Penzberg, Germany).

### 2.10. Statistics

All experiments were performed in triplicate. Statistical analysis was performed using GraphPad Prism 10.5.0 (GraphPad Software, Boston, MA, USA). Results are presented as mean ± standard deviation (SD). Normal distribution was tested using the Kolmogorow–Smirnow test. Comparisons were made using Student’s *t*-test for normally distributed data, or the Mann–Whitney U test for nonparametric data. A *p*-value ≤ 0.05 was considered statistically significant.

### 2.11. Use of AI-Assisted Editing

The authors used ChatGPT (GTP-5, OpenAI, San Francisco, CA, USA) for language polishing and editorial improvement of the manuscript text. All scientific content was created and verified by the authors.

## 3. Results

### 3.1. Static CytoSorb Adsorption

CytoSorb significantly (*p* ≤ 0.05) reduced the level of 6-MP (33 ± 11%), CsA (36 ± 9%), MPR (91 ± 2%), and MPA (37 ± 6%) in plasma. In parallel, 36 ± 6% of added 6-MP was absorbed by blood cells (Figure 2). TAC was not affected by CytoSorb but was reduced by blood cells (−57 ± 8%).

### 3.2. Protein Binding of Immunosuppressants

Among all tested immunosuppressants, 6-MP had the largest unbound fraction (Table 2). All other immunosuppressants showed >90% protein binding. Protein binding affects drug efficacy and dialyzability, with high protein-bound drugs being less efficiently removed via dialysis [5,19,20].

### 3.3. Dynamic CytoSorb Adsorption

In a dynamic ex vivo setup using a scaled-down CytoSorb cartridge, four of five tested immunosuppressants were depleted from human blood. The highest adsorption was observed for MPR (92 ± 3%), followed by MPA (76 ± 2%), 6-MP (47 ± 16%), and CsA (33 ± 11%). TAC showed only minimal and statistically non-significant removal (18 ± 16%), likely due to strong plasma protein binding. Control circuits confirmed that the reductions observed were attributable to CytoSorb treatment rather than cellular uptake (Figure 3).

### 3.4. Removal of Immunosuppressants by Dialysis

The dialyzability of immunosuppressants depends on molecular size, degree of protein binding, and the properties of the dialysis membrane [26,27]. Chronic dialysis achieved higher clearance rates for immunosuppressants than acute dialysis, primarily due to higher dialysate flow. 6-MP was efficiently removed in both settings, with clearance comparable to that of urea. MPR showed moderate removal in acute dialysis and high removal in chronic dialysis. MPA was only slightly reduced under chronic dialysis. Tacrolimus and cyclosporine A, both highly protein-bound, were not effectively removed by either dialysis type (Figure 4).

### 3.5. Clearance of Immunosuppressants in Extracorporeal Circuits

The clearance of immunosuppressants was calculated for CytoSorb treatment and for acute and chronic dialysis according to Equation (2) (Figure 5). CytoSorb showed the highest initial clearance values, with rapid adsorption during the first 30 min followed by a gradual decline due to adsorber saturation. At 15 min, mean clearance rates reached approximately 44 mL/min for mycophenolate, 49 mL/min for methylprednisolone, and 31 mL/min for 6-mercaptopurine, while tacrolimus and cyclosporine showed lower values around 15 mL/min and 2 mL/min, respectively.

In contrast, both dialysis systems exhibited markedly lower clearances. Acute dialysis reached up to 30 mL/min for 6-mercaptopurine and 28 mL/min for methylprednisolone, but values declined below 10 mL/min after 8 h. Chronic dialysis produced similar trends, with initially higher values for 6-mercaptopurine (~190 mL/min) and methylprednisolone (~75 mL/min) that decreased over time, whereas cyclosporine and tacrolimus remained below 2 mL/min. Overall, CytoSorb demonstrated a pronounced adsorption-driven clearance, while dialysis systems showed only limited elimination, mainly affecting small or weakly protein-bound compounds.

### 3.6. Whole Blood Cell Model

To assess the impact of CytoSorb on the concentration and immunomodulatory function of selected immunosuppressants in human whole blood, a dynamic adsorption experiment was conducted (Appendix A). Following overnight incubation with immunosuppressants, blood samples were subsequently treated with CytoSorb. Serum analysis revealed a significant reduction (*p* < 0.05) in the levels of 6-MP, CsA, MPR, and MPA following CytoSorb treatment (Figure 6). TAC, in contrast, showed no significant adsorption, confirming previous findings from static adsorption tests. For 6-MP and MPR, cellular uptake by blood cells contributed more to serum level reduction than CytoSorb adsorption alone.

To evaluate potential effects on immune function, blood samples were stimulated with LPS and anti-CD3 antibodies. Cytokine measurements indicated no significant differences in the secretion of TNF-α, IL-1β, IL-6, or IL-8 between CytoSorb-treated and untreated samples (Figure 7). Across all conditions, high donor variability was observed, as reflected by the relatively large standard deviations in cytokine levels (Appendix A). The small donor cohort (*n* = 3) represents a limitation of this ex vivo model. However, considerable inter-individual variability was observed, consistent with known differences in cytokine responsiveness among healthy donors.

## 4. Discussion

This study demonstrates that CytoSorb hemoadsorption can significantly reduce circulating concentrations of immunosuppressants under ex vivo conditions, with removal largely depending on the drug’s physicochemical properties. While this ex vivo approach allowed precise control of experimental parameters, it does not capture systemic pharmacokinetics, drug metabolism, or redistribution occurring in vivo. Therefore, translational relevance is limited and should be validated in clinical settings. These findings are of particular relevance in clinical contexts where extracorporeal blood purification is applied in patients under immunosuppressive therapy, such as transplant recipients or those suffering from autoimmune disorders [28]. While the in vitro and ex vivo performance of CytoSorb in removing inflammatory mediators and certain drugs appears promising, it must be critically acknowledged that clinical outcome data are less conclusive. Major randomized trials in septic shock and COVID-19 patients have not shown a survival benefit [8,9,10], and in some cases have even suggested potential harm. Supady et al. [11] stress the need for rigorous evaluation of hemoadsorption and advise against its routine clinical use outside controlled trials. This cautious interpretation is also reflected in the most recent S3 sepsis guideline (2025), which explicitly recommends that non-selective adsorption devices should not be used routinely in patients with sepsis or septic shock [12]. This underlines the limited evidence for patient-centered benefit and supports the view that such therapies should remain restricted to controlled clinical studies until further randomized data become available. Our findings, while relevant from a pharmacokinetic and mechanistic standpoint, should thus be interpreted within the context of this ongoing clinical uncertainty.

Highly protein-bound drugs, such as TAC and CsA showed limited removal by both dialysis and CytoSorb treatment. This observation is consistent with pharmacokinetic principles stating that only the unbound fraction of a drug is available for extracorporeal elimination [29,30,31,32]. In particular, TAC, which binds extensively to plasma proteins such as albumin and α1-acid glycoprotein [33], was not significantly removed, supporting the assumption that the drug–protein complex exceeds the molecular cut-off of CytoSorb’s porous polymer matrix [34].

In contrast, drugs such as MPR and MPA, which display lower protein binding and greater hydrophilicity, were efficiently cleared by CytoSorb. These findings align with previous studies indicating that substances with low molecular weight and minimal protein binding are optimal candidates for hemoadsorptive therapies [35,36,37]. 6-MP presented a unique profile: while moderately removed by CytoSorb, its plasma concentration was substantially reduced through cellular uptake. This is in line with its mechanism of action as a purine antimetabolite, which incorporates into nucleic acids and accumulates intracellularly [38]. These findings highlight the limitations of using plasma concentrations alone to predict pharmacodynamic outcomes, particularly for cytotoxic or antiproliferative agents. It should be noted that adsorption by CytoSorb is a time-dependent process, and progressive saturation of the polymer matrix may lead to a gradual decline in adsorption capacity during treatment. This phenomenon has been described previously in both clinical and experimental settings [34,39]. In our ex vivo setup, adsorption was monitored over a period of four hours, which likely captured the main adsorption phase but may not fully reflect long-term saturation kinetics observed in prolonged clinical use. The strong correlation observed between drug removal and protein binding also held true for dialysis. In accordance with prior reports [40], dialysis proved effective for the removal of unbound, low-molecular compounds such as 6-MP, while agents such as TAC and CsA were poorly dialyzable due to their high protein affinity and low free fractions in plasma. In addition to protein binding, the clinical relevance of extracorporeal drug removal depends on the compound’s volume of distribution (Vd). Drugs with a large Vd are predominantly located in peripheral tissues, leaving only a small fraction in the circulating plasma available for adsorption. Consequently, the observed reductions mainly reflect depletion of the intravascular compartment rather than total body clearance. In this context, methylprednisolone (Vd ≈ 1.4 L/kg) [23] and mycophenolate (Vd ≈ 3.6 L/kg) [24] exhibit higher distribution volumes compared to 6-mercaptopurine (Vd ≈ 0.9 L/kg) [41], which may explain the limited clinical relevance of CytoSorb adsorption for some agents despite measurable ex vivo removal. Similar pharmacokinetic principles have been emphasized in standard pharmacology references and underline that significant systemic drug depletion by CytoSorb is expected mainly for substances with low protein binding and low Vd. However, these clearance values represent adsorption rates within a closed ex vivo circuit and should not be interpreted as true pharmacokinetic parameters. Drug distribution into peripheral compartments and metabolic turnover were not modeled in this setup.

Despite the observed reduction in plasma concentrations of selected immunosuppressants following CytoSorb treatment, no significant differences in the release of IL-1β, IL-6, IL-8, or TNF-α were detected in an ex vivo whole blood model. While this may suggest preserved immune responsiveness regarding these specific pathways, the limited cytokine panel and the artificial nature of the model preclude broader conclusions about the overall immunosuppressive state. Therefore, our findings should be interpreted as indicative of preserved short-term cytokine response under defined ex vivo conditions, rather than a comprehensive assessment of immunosuppressive function. The cytokine measurements were limited to an acute 4 h window post-stimulation and thus cannot reflect long-term immunological processes such as lymphocyte proliferation, antigen presentation, or feedback regulation. Future studies should extend the observation period to capture delayed effects. One possible explanation is the rapid onset of action of calcineurin inhibitors like TAC and CsA [42], which may exert their effects before removal. In contrast, drugs like 6-MP require prolonged exposure—often several weeks—to achieve their full immunosuppressive effects [43], a process that cannot be replicated in short-term ex vivo models. It should be noted that the LPS and anti-CD3 stimulation assays used in this study represent a simplified ex vivo approach that cannot fully reproduce the complex, time-dependent mechanisms of immunosuppression in vivo. Nevertheless, these assays provide qualitative evidence that the short-term reduction in selected immunosuppressants by CytoSorb did not lead to measurable alterations in cytokine release compared to untreated controls. This suggests that CytoSorb treatment under the tested conditions does not acutely impair the cytokine response capacity of circulating immune cells, although long-term or cumulative effects cannot be excluded.

These findings indicate that CytoSorb can reduce plasma concentrations of selected immunosuppressants under defined ex vivo conditions. Due to the complex and predominantly cell-mediated nature of immunosuppressive mechanisms in vivo, and the limited scope of our cytokine panel, no conclusions regarding the safety or functional preservation of immunosuppression can be drawn from this study. Clinical use should therefore be guided by therapeutic drug monitoring and further validated in prospective in vivo trials. However, the cumulative effects of repeated treatments, particularly in critically ill patients with altered pharmacokinetics, remain unclear and require further clinical investigation [44]. Beyond these mechanistic considerations, the clinical context is equally important. Clinically, these interactions are particularly relevant in transplant recipients who may experience peri-operative cytokine storm or acute rejection risk. In such scenarios, CytoSorb has been considered as an adjunct to stabilize hemodynamics or reduce inflammatory load. Our data suggest that while CytoSorb can lower circulating drug concentrations, short-term use is unlikely to abolish immunosuppressive efficacy. Nevertheless, careful therapeutic drug monitoring is recommended in these high-risk situations. From a clinical perspective, the observed adsorption of certain immunosuppressants underscores the importance of therapeutic drug monitoring during CytoSorb treatment, particularly for compounds with low protein binding and small to moderate distribution volumes. Nevertheless, due to the ex vivo nature of this study, no dosing recommendations can be made. Future clinical trials should include pharmacokinetic measurements before and after hemoadsorption to determine the potential need for dose adjustment and to establish evidence-based guidance for patients receiving immunosuppressive therapy during CytoSorb treatment.

In summary, CytoSorb selectively depletes immunosuppressants according to their molecular properties, without detectable impairment of immune responsiveness in this short-term ex vivo model. A single short-term CytoSorb treatment reduced immunosuppressant drug levels but did not significantly affect the levels of IL-1β, IL-6, IL-8, or TNF-α in our ex vivo whole blood model. As this setup does not fully recapitulate the complexity of the in vivo immune response, the translational relevance of this observation remains limited and requires further investigation. Individualized immunosuppressant management should rely on therapeutic drug monitoring and validated clinical studies. Our findings do not permit conclusions on whether the pharmacodynamic effects of immunosuppressants are maintained following CytoSorb treatment, as critical immunological processes—such as T-cell proliferation, antigen presentation, and long-term suppression—were not assessed in this model.

## 5. Conclusions

In this ex vivo study, CytoSorb treatment led to a measurable reduction in plasma levels of CsA, 6-MP, MPR, and MPA. While our results confirm that CytoSorb can reduce plasma concentrations of several immunosuppressants ex vivo, the complexity of immunosuppressive therapy in clinical practice precludes any recommendation regarding safety or dose adjustment based on this study alone.

## Figures and Tables

**Figure 1 pharmaceutics-17-01468-f001:**
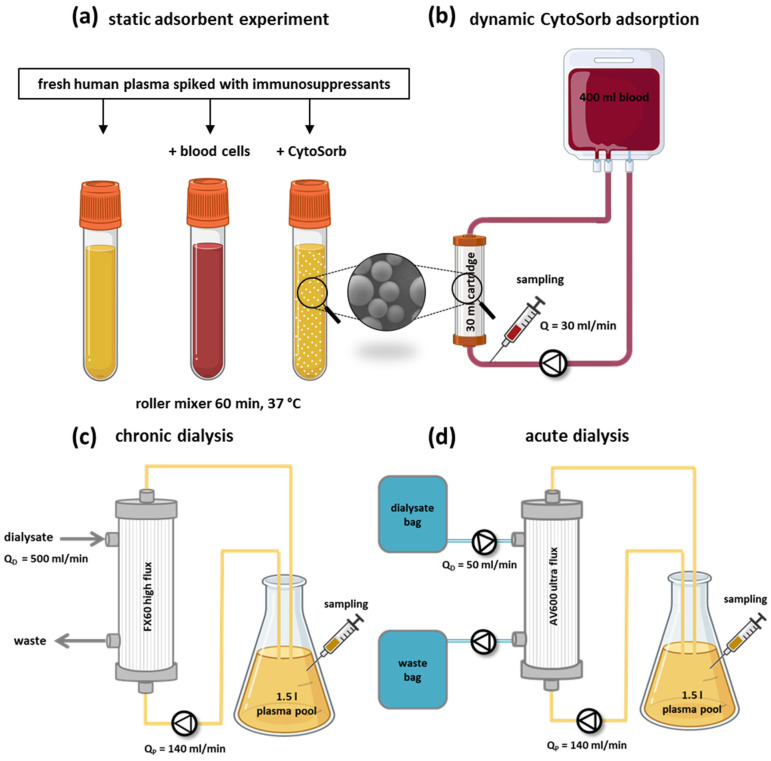
(**a**) Graphical representation of the static CytoSorb adsorption setup used to determine the depletion of immunosuppressants from fresh human plasma. (**b**) Downscaled CytoSorb circulation circuits, in which 400 mL of fresh human blood spiked with immunosuppressants was circulated for 4 h through a 30 mL CytoSorb-filled or empty cartridge (control circuit) at a blood flow rate of 30 mL/min. Samples were taken to determine the clearance of each immunosuppressant. In vitro hemodialysis setups used to assess immunosuppressant removal in chronic (**c**) and acute (**d**) dialysis. Chronic dialysis was performed using a Fresenius 4008H device with an FX60 high-flux filter and calcium-free dialysate at a dialysate flow rate of Q_D = 500 mL/min, while a 1.5 L plasma pool was circulated at Q_P = 140 mL/min for 4 h. Acute dialysis was performed using a multiFiltrate device with an AV600 ultra-flux filter and calcium-free dialysate at Q_D = 50 mL/miand a plasma pool of 1.5 L at Q_P = 140 mL/min for 8 h.

**Figure 2 pharmaceutics-17-01468-f002:**
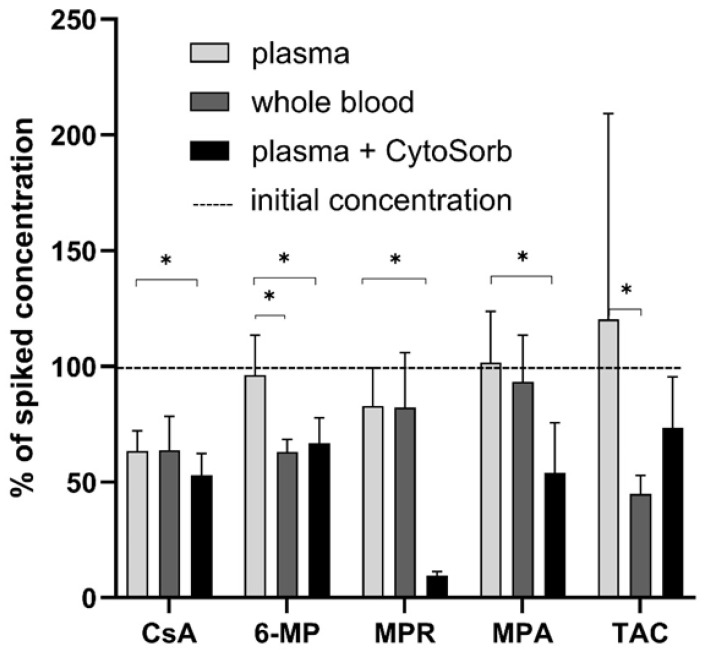
Depletion of immunosuppressants by CytoSorb and blood cells (*n* = 3). Significant differences (*p* ≤ 0.05) are marked with *.

**Figure 3 pharmaceutics-17-01468-f003:**
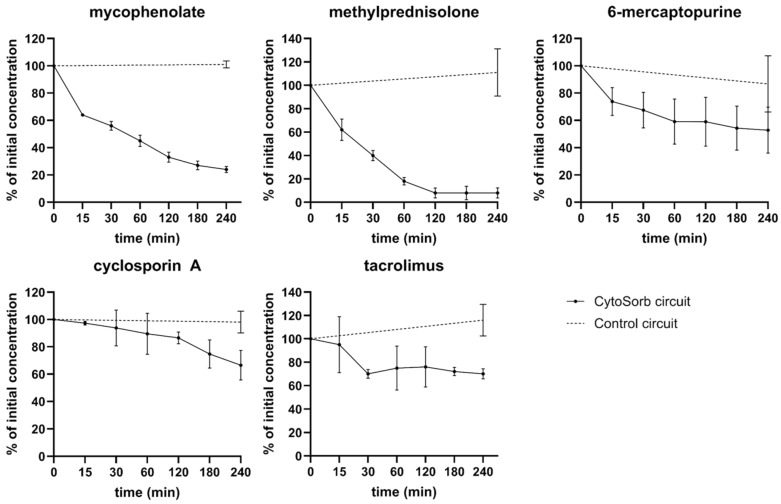
Immunosuppressant removal during CytoSorb treatment in an ex vivo model.

**Figure 4 pharmaceutics-17-01468-f004:**
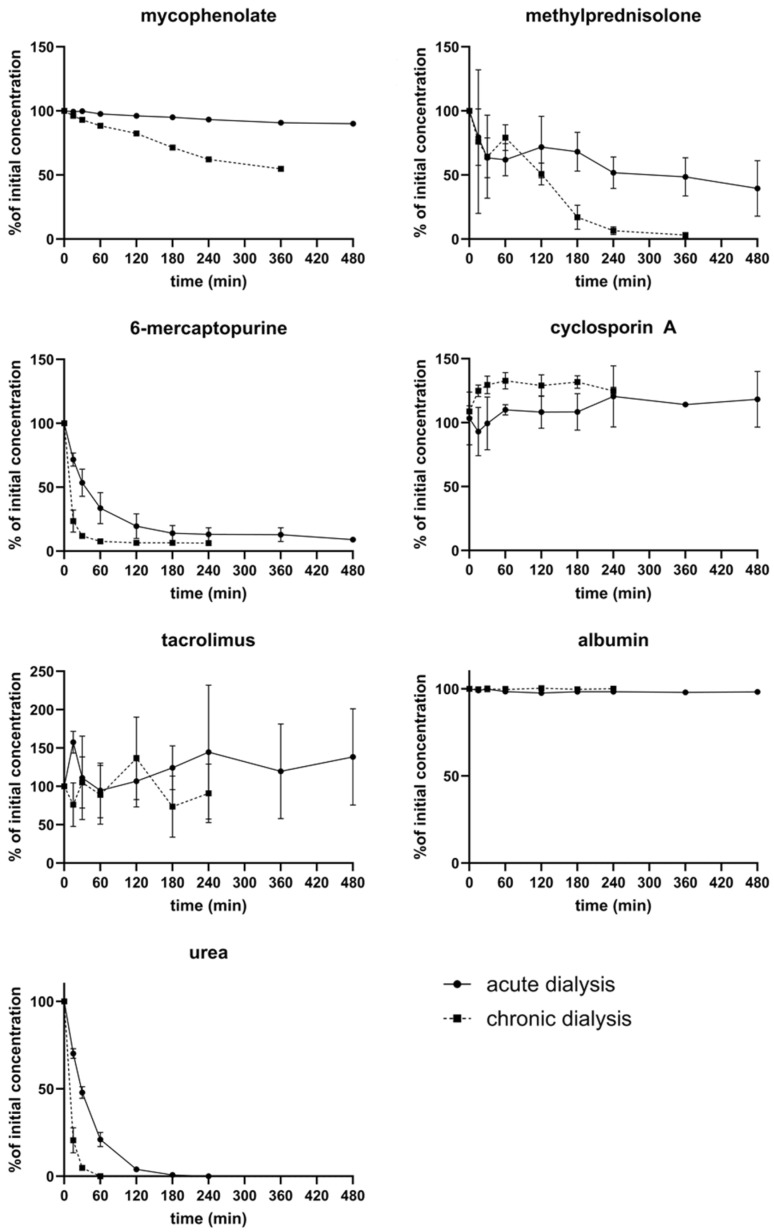
Depletion of individual immunosuppressants by ex vivo dialysis.

**Figure 5 pharmaceutics-17-01468-f005:**
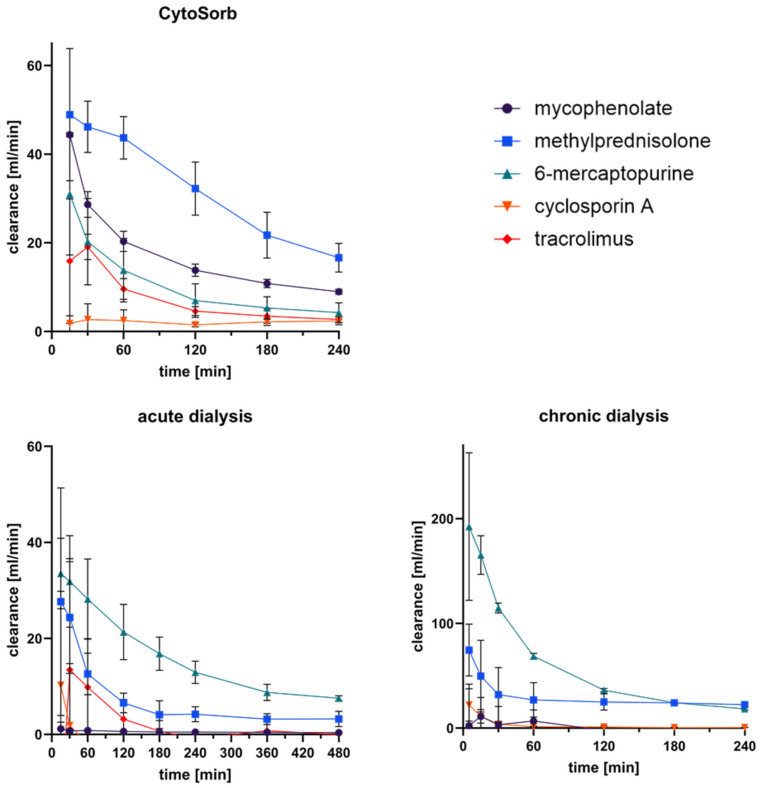
Calculated clearance of immunosuppressants during CytoSorb treatment, acute dialysis, and chronic dialysis. Time-dependent clearance (mL/min) of mycophenolate, methylprednisolone, 6-mercaptopurine, cyclosporine A, and tacrolimus was determined using Equation (2).

**Figure 6 pharmaceutics-17-01468-f006:**
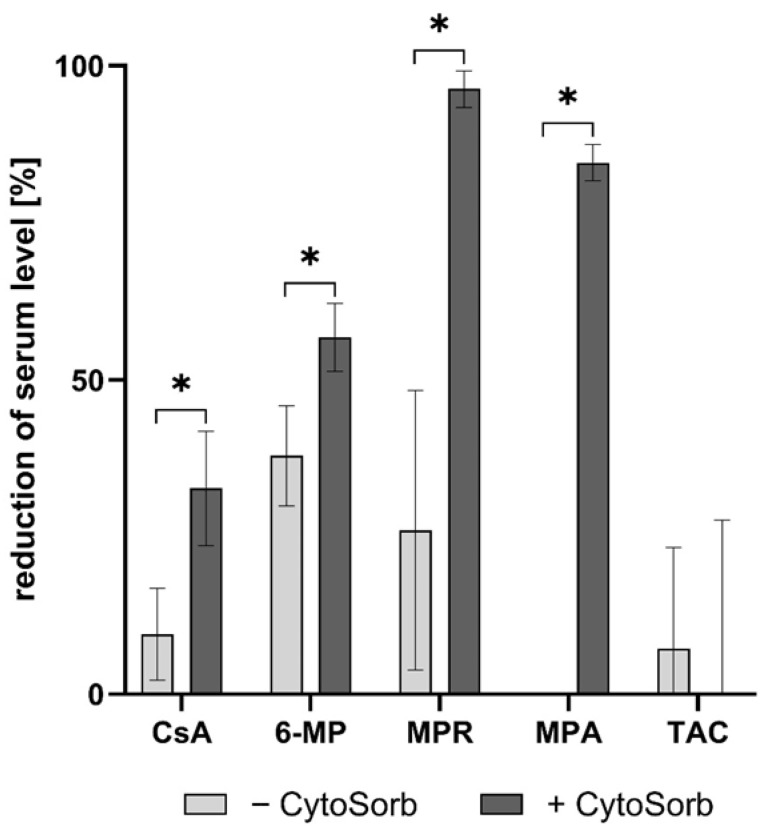
Reduction in immunosuppressant serum levels after CytoSorb treatment in a whole blood model (*n* = 6). Following overnight incubation with immunosuppressants, aliquots of the blood were either treated with CytoSorb (+CytoSorb) or left untreated (−CytoSorb). Data are presented as the percentage reduction relative to untreated controls. Significant differences (*p* < 0.05) are indicated by *.

**Figure 7 pharmaceutics-17-01468-f007:**
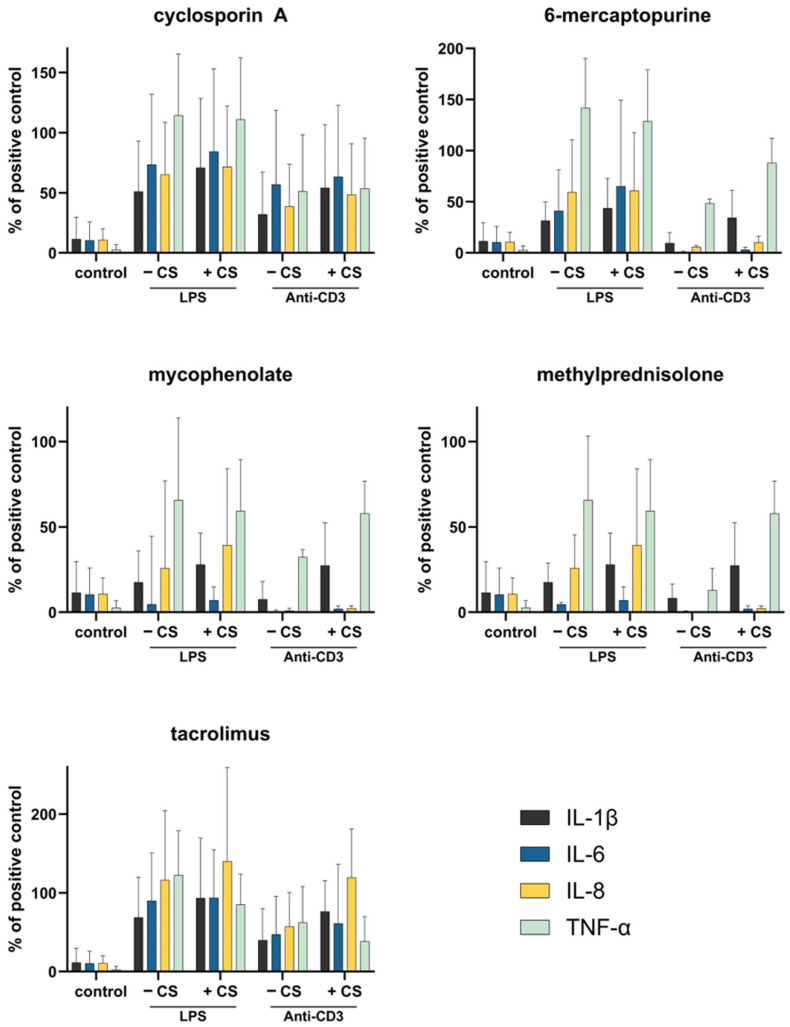
Effect of CytoSorb treatment on the immunosuppressive capacity in a whole blood cell model. Following CytoSorb treatment, blood samples were stimulated with LPS (*E. coli*) and anti-CD3 to assess the activation capability of leukocytes. After 4 h of incubation, cytokine secretion was compared between CytoSorb-treated (+CS) and untreated (−CS) samples. Data represent mean cytokine serum levels from three independent experiments (*n* = 3). The positive control consisted of LPS and anti-CD3-stimulated blood from the same donor without immunosuppressants and was set to 100%. Unstimulated blood without immunosuppressants served as negative control.

**Table 1 pharmaceutics-17-01468-t001:** Spiked plasma levels of the immunosuppressants tested in this ex vivo study.

Immunosuppressant	Plasma Spike-Concentration [ng/mL]	Immunosuppressive Action
tacrolimus (TAC)	20	TAC inhibits calcineurin and consequently IL-2 synthesis and thus T cell proliferation.
mycophenolate (MPA)	4000	MPA inhibits the synthesis of guanosine and thus the proliferation of T and B lymphocytes.
methylprednisolone (MPR)	200	MPR is a synthetic glucocorticoid and has anti-inflammatory and immunosuppressive effects.
cyclosporin (CsA)	1200	CsA inhibits calcineurin and consequently IL-2 synthesis and thus T cell proliferation.
6-mercaptopurine (6-MP)	1000	6-MP is an antimetabolite, which means that it is incorporated into the DNA instead of adenine and guanine during cell division. The resulting DNA thereby loses its function.

**Table 2 pharmaceutics-17-01468-t002:** Results of the protein binding study compared to protein binding data from the literature.

	Protein Binding in Plasma [%]	
	Current Study	Reference
6-MP	33.8 ± 6.3	39 [21]
CsA	99.7 ± 1.7	90 [22]
MPR	93.4 ± 6.5	77 [23]
MPA	97.9 ± 0.3	97 [24]
TAC	97.9 ± 10.9	99 [25]

## Data Availability

The data presented in this study are available on reasonable request from the corresponding author.

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
