# Peer review of "Does CytoSorb Interfere with Immunosuppression? A Pharmacokinetic and Functional Evaluation"

_pharmaceutics, 2025, doi:10.3390/pharmaceutics17111468_

Round 1

Reviewer 1 Report

Comments and Suggestions for Authors

Thank you for giving me the opportunity to review this paper.
I find the aim of the study and its contents very relevant.
However, I believe that the presentation of the data needs to be thoroughly revised in order to make it more accessible to readers.

Methods:

  • I would suggest helping the reader with images to clarify what is meant by static Cytosorb adsorption versus dynamic Cytosorb adsorption.

  • The characteristics of the biological fluids used and the related calculations should be included in the same paragraph.

  • I find it of little use to include both chronic dialysis and acute dialysis. In any case, I recommend summarizing the two paragraphs, highlighting the main technical differences.

  • I consider the incubation of the biological fluid with drugs prior to treatment with Cytosorb to be a process with very limited in vivo applicability. The measurement taken before the insertion of the Cytosorb is in fact sufficient on its own to establish a baseline of the values, also accounting for cellular metabolism. I would therefore remove this part, as it only complicates the interpretation of the results.

  • Finally, the measurement using an LPS stimulation test to evaluate the impact of removing immunosuppressants from the cartridge seems extremely reductive. I would not include this data.

Results

The results are also not presented clearly.

  • Report clearance in ml/min instead of percentage in the time diagram.

  • Convert the percentages into histograms with and without Cytosorb (totals over the observation period).

  • I would eliminate the table on clearance and protein binding, as it adds no new information. It is sufficient to mention this in the text.

For now, I am indicating these revisions. I would like to review the manuscript again after a radical revision of the methods and results, in order to make the study’s findings more accessible.

Author Response

Comment 1: I would suggest helping the reader with images to clarify what is meant by static Cytosorb adsorption versus dynamic Cytosorb adsorption.

Response 1: We thank the reviewer for this valuable suggestion. To address this, we have ensured that the static and dynamic CytoSorb adsorption setups are clearly illustrated in Figure 1. The static adsorption experiment is now explicitly mentioned in the figure legend to help the reader distinguish between both approaches. The characteristics of the biological fluids used and the related calculations should be included in the same paragraph.

Comment 2: I find it of little use to include both chronic dialysis and acute dialysis. In any case, I recommend summarizing the two paragraphs, highlighting the main technical differences.

Response 2: We appreciate this helpful comment. We agree that the previous version described both dialysis setups in too much detail. In the revised manuscript, the two paragraphs have been merged into a single concise section (“Dialysis setups”), summarizing the key technical differences (flow rates, dialyzer type, and system used) while removing redundant text. This change improves readability and better highlights the rationale for comparing high-flow chronic versus low-flow acute dialysis conditions.

Comment 3: I consider the incubation of the biological fluid with drugs prior to treatment with Cytosorb to be a process with very limited in vivo applicability. The measurement taken before the insertion of the Cytosorb is in fact sufficient on its own to establish a baseline of the values, also accounting for cellular metabolism. I would therefore remove this part, as it only complicates the interpretation of the results.

Response 3: The overnight pre-incubation step was removed from the main workflow and is now described in the Supplementary Information, as its in vivo relevance is limited. All baseline measurements prior to CytoSorb insertion were used as reference values.

Comment 4: Finally, the measurement using an LPS stimulation test to evaluate the impact of removing immunosuppressants from the cartridge seems extremely reductive. I would not include this data.

Response 4: We thank the reviewer for this valuable comment. We agree that the LPS and anti-CD3 stimulation assays represent a simplified ex vivo approach that cannot capture the full complexity of immunosuppressive mechanisms in vivo. Our intention, however, was not to model clinical immunosuppression, but rather to demonstrate that CytoSorb treatment under the tested conditions does not acutely abolish cytokine release capacity in whole blood. We have clarified this limitation in the revised Discussion section and emphasized that these data serve as a qualitative indicator of preserved short-term immune responsiveness only. We therefore kindly ask to keep these data in the manuscript, as they provide supportive evidence that the short-term reduction of selected immunosuppressants by CytoSorb did not result in detectable changes in the cytokine release profile when compared to non-CytoSorb-treated samples.

In the discussion part, following text was added: “It should be noted that the LPS and anti-CD3 stimulation assays used in this study represent a simplified ex vivo approach that cannot fully reproduce the complex, time-dependent mechanisms of immunosuppression in vivo. Nevertheless, these assays provide qualitative evidence that the short-term reduction of selected immunosuppressants by CytoSorb did not lead to measurable alterations in cytokine release compared to untreated controls. This suggests that CytoSorb treatment under the tested conditions does not acutely impair the cytokine response capacity of circulating immune cells, although long-term or cumulative effects cannot be excluded.”

Comment 5: Report clearance in ml/min instead of percentage in the time diagram.

Response 5: We thank the reviewer for this valuable suggestion. We have now implemented an additional figure (new Figure 5) in which the calculated clearance (mL/min) for each immunosuppressant is plotted over time. Clearance values were derived using Equation (2) from the concentration–time profiles for all three treatment modalities (CytoSorb hemoadsorption, acute dialysis, and chronic dialysis). This modification provides a more quantitative and pharmacokinetically meaningful representation of the elimination kinetics.

Comment 6: Convert the percentages into histograms with and without Cytosorb (totals over the observation period).

Response 6: We thank the reviewer for this helpful suggestion. In the revised version, we have converted the results from depletion of immunosuppressivea from the blood model into a histogram (now Figure 6) illustrating the percentage reduction of immunosuppressant serum levels with and without CytoSorb treatment (+/- CytoSorb).

Comment 7: I would eliminate the table on clearance and protein binding, as it adds no new information. It is sufficient to mention this in the text.

Response 7: We thank the reviewer for this comment. The table summarizing clearance and protein binding data has been removed as suggested. The relevant information is now described directly in the text for better readability and conciseness.

For your convenience, all modifications and newly added text in the revised manuscript have been highlighted in red to facilitate the review of changes.

Reviewer 2 Report

Comments and Suggestions for Authors

Introduction: 

  • EUPHRATES and ABDOMIX Trial investigated the use of Polymyxin-B, which is totally different to cytokine adsorption
  • as Cytosorb is often used in europe, you might cite the new S3 Sepsis guideline, where adsorber should not be used in patients with sepsis 
  • Ref 16 reports not a patient with liver/heart transplant. However, there a different studies available reporting a beneficial clinical outcome in patients with heart transplant 

Results: 

- Remove: This section may be divided by subheadings. It should provide a concise and precise description of the 253
experimental results, their interpretation, as well as the experimental conclusions that can be drawn. 254

- Is there a time-dependent adsorption. There are different publications that report a saturation kinetic during therapy 

- What is the volume of distribution for the different drugs? Is the adsorption i.e. with a volume of distribution of 50L clinically relevant?

Discussion: 

  • The references 17 to 31 in the reference list do not refer to the references named in the discussion.
  • The clinical implication is missing. Should the dose be adjusted? If so, by how much? Should a clinical study be conducted with measurements before and after the adsorber and in the blood?  

Author Response

Comment 1: EUPHRATES and ABDOMIX Trial investigated the use of Polymyxin-B, which is totally different to cytokine adsorption

Response 1: We thank the reviewer for this important clarification. We fully agree that both the EUPHRATES and ABDOMIX trials evaluated polymyxin-B hemoperfusion, which targets endotoxin removal rather than broad-spectrum cytokine adsorption. We have therefore revised the Introduction accordingly to avoid potential confusion. The revised text now distinguishes between endotoxin adsorption (polymyxin-B) and cytokine adsorption (CytoSorb), while retaining the reference to these trials in the broader context of extracorporeal blood purification strategies that have not yet translated into consistent survival benefits.

Original sentence: “Several randomized controlled trials, including EUPHRATES [8], ABDOMIX [9], and CYCOV [10], have failed to demonstrate a survival benefit.”

Revised version: “Several randomized controlled trials on extracorporeal blood purification approaches—including endotoxin removal using polymyxin-B hemoperfusion (EUPHRATES [8] and ABDOMIX [9]) and cytokine adsorption using CytoSorb (CYCOV [10])—have failed to demonstrate a survival benefit.”

Comment 2: as Cytosorb is often used in europe, you might cite the new S3 Sepsis guideline, where adsorber should not be used in patients with sepsis 

Response 2: We thank the reviewer for drawing attention to the updated S3 Sepsis guideline (2025), which indeed includes a recommendation against the use of non-selective adsorption devices (i.e. adsorbers) in patients with sepsis and septic shock.

In the revised manuscript, we have incorporated this guideline statement into the Introduction and Discussion to contextualize our work more accurately. We now explicitly state that current guidelines do not recommend routine use of adsorbers outside of clinical trials, reflecting the uncertainty and lack of evidence for survival benefit.

We added in the introduction part: “Furthermore, the most recent S3 guideline for sepsis (2025) explicitly recommends that non-selective adsorption devices should not be used routinely in patients with sepsis or septic shock [12]. Thus, while CytoSorb and similar devices have been applied in Europe in clinical practice and in investigational settings, their routine use is not supported by guideline consensus.”

We added in the discussion part: “This cautious interpretation is also reflected in the most recent S3 sepsis guideline (2025), which explicitly recommends that non-selective adsorption devices should not be used routinely in patients with sepsis or septic shock [12]. This underlines the limited evidence for patient-centered benefit and supports the view that such therapies should remain restricted to controlled clinical studies until further randomized data become available.”

Comment 3: Ref 16 reports not a patient with liver/heart transplant. However, there a different studies available reporting a beneficial clinical outcome in patients with heart transplant 

Response 3: We thank the reviewer for this helpful correction. We agree that Ref. 16 (Träger et al., 2016) does not report a liver or heart transplant case but rather describes cytokine adsorption in a patient with severe ARDS and multiple organ failure. We have therefore revised this section accordingly and replaced this reference with appropriate clinical studies reporting the use of CytoSorb in heart transplant recipients showing beneficial perioperative outcomes (e.g., Nemeth et al., Clin Transplant 2018; Tomescu et al., Int J Artif Organs 2016).

Results: 

Comment 4: Remove: This section may be divided by subheadings. It should provide a concise and precise description of the 253
experimental results, their interpretation, as well as the experimental conclusions that can be drawn. 254

Response 4: We thank the reviewer for noting this formatting error. The indicated placeholder text was unintentionally included in the previous version and has been removed from the revised manuscript.

Comment 5: Is there a time-dependent adsorption. There are different publications that report a saturation kinetic during therapy 

Response 5: We thank the reviewer for this relevant comment. Indeed, previous studies have reported a time-dependent decline in adsorption efficiency during CytoSorb therapy due to progressive saturation of binding sites on the polymer surface. In our ex vivo setup, adsorption was monitored over a period of up to 240 minutes, and the calculated clearances (Figure 5) represent the mean adsorption rate over time. While a gradual reduction in clearance can be observed for some compounds, our study was not designed to quantitatively determine saturation kinetics. We have clarified this limitation in the revised Discussion section.

We added following text in the discussion part: “It should be noted that adsorption by CytoSorb is a time-dependent process, and progressive saturation of the polymer matrix may lead to a gradual decline in adsorption capacity during treatment. This phenomenon has been described previously in both clinical and experimental settings [34,39]. In our ex vivo setup, adsorption was monitored over a period of four hours, which likely captured the main adsorption phase but may not fully reflect long-term saturation kinetics observed in prolonged clinical use.”

Comment 6: What is the volume of distribution for the different drugs? Is the adsorption i.e. with a volume of distribution of 50L clinically relevant?

Response 6: We thank the reviewer for this valuable pharmacokinetic comment. The clinical relevance of extracorporeal drug removal indeed depends on both protein binding and the volume of distribution (Vd). Drugs with a high Vd are mainly distributed in peripheral tissues and therefore less susceptible to extracorporeal elimination. In our study, methylprednisolone (Vd ≈ 1.4 L/kg) and mycophenolate (Vd ≈ 3.6 L/kg) have relatively high Vd values, whereas 6-mercaptopurine (Vd ≈ 0.9 L/kg) is lower. We have included this consideration in the revised Discussion section to clarify that, despite measurable ex vivo adsorption, the in vivo relevance is likely limited for drugs with large distribution volumes.

We added following text in the discussion part: In addition to protein binding, the clinical relevance of extracorporeal drug removal depends on the compound’s volume of distribution (Vd). Drugs with a large Vd are predominantly located in peripheral tissues, leaving only a small fraction in the circulating plasma available for adsorption. Consequently, the observed reductions mainly reflect depletion of the intravascular compartment rather than total body clearance. In this context, methylprednisolone (Vd 1.4 L/kg) [23] and mycophenolate (Vd 3.6 L/kg) [24] exhibit higher distribution volumes compared to 6-mercaptopurine (Vd 0.9 L/kg) [41], which may explain the limited clinical relevance of CytoSorb adsorption for some agents despite measurable ex vivo removal. Similar pharmacokinetic principles have been emphasized in standard pharmacology references and underline that significant systemic drug depletion by CytoSorb is expected mainly for substances with low protein binding and low Vd.”

Discussion: 

Comment 7: The references 17 to 31 in the reference list do not refer to the references named in the discussion.

Response 7: We thank the reviewer for carefully checking the reference list. We have thoroughly revised and corrected all in-text citations to ensure that reference numbering and corresponding entries in the reference list are consistent throughout the manuscript. The numbering of references 17–31 has been updated accordingly in the revised version.

Comment 8: The clinical implication is missing. Should the dose be adjusted? If so, by how much? Should a clinical study be conducted with measurements before and after the adsorber and in the blood?  

Response 8: We thank the reviewer for this valuable comment. The present study was designed as an ex vivo proof-of-concept to assess the adsorption of selected immunosuppressants by CytoSorb under controlled conditions. Therefore, no direct dosing recommendations can be derived. However, our findings indicate that substances with low protein binding and small to moderate distribution volumes (e.g., 6-mercaptopurine, methylprednisolone) are susceptible to adsorption and may require therapeutic drug monitoring if CytoSorb is applied clinically. We have emphasized this point in the revised Discussion section and agree that future clinical studies should include pharmacokinetic measurements before and after hemoadsorption to define potential dose adjustment strategies.

We added following text passage in the discussion part: “From a clinical perspective, the observed adsorption of certain immunosuppressants underscores the importance of therapeutic drug monitoring during CytoSorb treatment, particularly for compounds with low protein binding and small to moderate distribution volumes. Nevertheless, due to the ex vivo nature of this study, no dosing recommendations can be made. Future clinical trials should include pharmacokinetic measurements before and after hemoadsorption to determine the potential need for dose adjustment and to establish evidence-based guidance for patients receiving immunosuppressive therapy during CytoSorb treatment.”

For your convenience, all modifications and newly added text in the revised manuscript have been highlighted in red to facilitate the review of changes!

Reviewer 3 Report

Comments and Suggestions for Authors

The manuscript titled -Does CytoSorb Interfere with Immunosuppression? A
Pharmacokinetic and Functional Evaluation by Harm et al. presents an ex vivo study and insights into how CYtosorb interacts with immunosuppressants. 
The concern of the manuscript is the number of limitations-
Translational relevance is weak. 
While the study is ex vivo, it cannot explain pharmacokinetics or drug distribution. 
The cytokine measurement time is short, 4 h post-stimulation, which lacks long-term outcome information on cell proliferation, antigen presentation, or chronic effects.
Although donor variability has been reported, only 3 donors were used, which limits the conclusion. 
Specific scenarios -transplant rejection risk, Cytosorb use in immunosuppressant patients, are missing.
Figures and data presentation should be improved. Figure 6 has no error bar. How many samples are there? 

Author Response

Comment 1: The concern of the manuscript is the number of limitations-
Translational relevance is weak. 

Response 1: We thank the reviewer for this important observation. We agree that the translational transfer of our ex vivo results to clinical pharmacokinetics is limited. Our intention was not to model in vivo pharmacokinetics but to provide mechanistic insight into adsorption patterns and short-term immunological effects under controlled conditions. We have emphasized this limitation in the Discussion and clarified that future in vivo studies are needed to confirm clinical relevance

Following text was added to the discussion part: “While this ex vivo approach allowed precise control of experimental parameters, it does not capture systemic pharmacokinetics, drug metabolism, or redistribution occurring in vivo. Therefore, translational relevance is limited and should be validated in clinical settings.”

Comment 2: While the study is ex vivo, it cannot explain pharmacokinetics or drug distribution. 

Response 2: We agree with the reviewer that our ex vivo model does not account for pharmacokinetic processes such as distribution, metabolism, or elimination beyond the extracorporeal circuit. We have clarified this point in the Discussion and emphasized that the observed clearance values reflect adsorption capacity rather than true pharmacokinetics.

Following text was added to the discussion part: “In addition to protein binding, the clinical relevance of extracorporeal drug removal depends on the com-pound’s volume of distribution (Vd). Drugs with a large Vd are predominantly located in peripheral tissues, leaving only a small fraction in the circulating plasma available for adsorption. Consequently, the observed reductions mainly reflect depletion of the intravascular compartment rather than total body clearance. In this context, methylprednisolone (Vd ≈ 1.4 L/kg) [23] and mycophenolate (Vd ≈ 3.6 L/kg) [24] exhibit higher distri-bution volumes compared to 6-mercaptopurine (Vd ≈ 0.9 L/kg) [41], which may explain the limited clinical relevance of CytoSorb adsorption for some agents despite measurable ex vivo removal. Similar pharmacoki-netic principles have been emphasized in standard pharmacology references and underline that significant systemic drug depletion by CytoSorb is expected mainly for substances with low protein binding and low Vd. However, these clearance values represent adsorption rates within a closed ex vivo circuit and should not be interpreted as true pharmacokinetic parameters. Drug distribution into peripheral compartments and meta-bolic turnover were not modeled in this setup.”

Comment 3: The cytokine measurement time is short, 4 h post-stimulation, which lacks long-term outcome information on cell proliferation, antigen presentation, or chronic effects.

Response 3: We thank the reviewer for this valid point. Indeed, cytokine release within 4 h provides only a short-term snapshot of immune activation and does not address longer-term effects such as T-cell proliferation or antigen presentation. We have now explicitly acknowledged this limitation in the Discussion and clarified that the assay was designed to detect acute cytokine responses only.

Following text was added to the discussion part: “The cytokine measurements were limited to an acute 4-hour window post-stimulation and thus cannot reflect long-term immunological processes such as lymphocyte proliferation, antigen presentation, or feedback regu-lation. Future studies should extend the observation period to capture delayed effects.”

Comment 4: Although donor variability has been reported, only 3 donors were used, which limits the conclusion.

Response 4: We agree that the limited number of donors represents a constraint. Due to the complexity of the ex vivo setup, experiments were restricted to three independent donors to ensure technical reproducibility. Nevertheless, inter-individual variability was evident in cytokine responses, which we now discuss explicitly.

Following text was added to the results part: “The small donor cohort (n = 3) represents a limitation of this ex vivo model. However, considerable inter-individual variability was observed, consistent with known differences in cytokine responsiveness among healthy donors.”

Comment 5: Specific scenarios -transplant rejection risk, Cytosorb use in immunosuppressant patients, are missing.

Response 5: We appreciate this suggestion and have added a short paragraph in the Discussion highlighting potential clinical contexts where CytoSorb could be relevant, such as peri-transplant cytokine storm or acute rejection risk in patients on immunosuppression.

Following text was added to the discussion part: “Clinically, these interactions are particularly relevant in transplant recipients who may experience pe-ri-operative cytokine storm or acute rejection risk. In such scenarios, CytoSorb has been considered as an ad-junct to stabilize hemodynamics or reduce inflammatory load. Our data suggest that while CytoSorb can lower circulating drug concentrations, short-term use is unlikely to abolish immunosuppressive efficacy. Neverthe-less, careful therapeutic drug monitoring is recommended in these high-risk situations. From a clinical per-spective, the observed adsorption of certain immunosuppressants underscores the importance of therapeutic drug monitoring during CytoSorb treatment, particularly for compounds with low protein binding and small to moderate distribution volumes. Nevertheless, due to the ex vivo nature of this study, no dosing recommen-dations can be made. Future clinical trials should include pharmacokinetic measurements before and after hemoadsorption to determine the potential need for dose adjustment and to establish evidence-based guidance for patients receiving immunosuppressive therapy during CytoSorb treatment.”

Comment 6: Figures and data presentation should be improved. Figure 6 has no error bar. How many samples are there? 

Response 6: We thank the reviewer for this observation. Figure 6 has been updated to include error bars representing the standard deviation. The number of samples (n = 6) has been added to the figure legend.

Round 2

Reviewer 1 Report

Comments and Suggestions for Authors

The authors have addressed all my questions. I now find the paper much more readable. I have no further requests.

Reviewer 2 Report

Comments and Suggestions for Authors

Thank you for the changes!

Reviewer 3 Report

Comments and Suggestions for Authors

Thank you for improving the manuscript. I have no further comment.